# Genomic Characterisation of Canis Familiaris Papillomavirus Type 24, a Novel Papillomavirus Associated with Extensive Pigmented Plaque Formation in a Pug Dog

**DOI:** 10.3390/v14112357

**Published:** 2022-10-26

**Authors:** John S. Munday, Kristene Gedye, Matthew A. Knox, Philippa Ravens, Xiaoxiao Lin

**Affiliations:** 1Pathobiology, School of Veterinary Science, Massey University, Palmerston North 4410, New Zealand; 2Molecular Epidemiology Laboratory, School of Veterinary Science, Massey University, Palmerston North 4410, New Zealand; 3Small Animal Specialist Hospital, North Ryde, NSW 2213, Australia; 4Massey Genome Service, Massey University, Palmerston North 4410, New Zealand

**Keywords:** dog, papillomavirus, viral plaque, pigmented plaque, canine papillomavirus (CPV), CPV24, neoplasia, skin

## Abstract

Numerous large dark plaques developed over the ventrum, legs and head of a 9-year-old pug dog over a 4-year-period. Histology confirmed a diagnosis of viral pigmented plaque and a short section of a novel papillomavirus (PV) type was amplified using consensus PCR primers. Taking advantage of the circular nature of PV DNA, ‘outward facing’ PCR primers allowed amplification of the full sequence. As this is the 24th PV known to infect dogs, the novel PV was designated canine papillomavirus (CPV) type 24. The CPV24 genome contained putative coding regions for 5 early proteins and 2 late ones. The CPV24 open reading frame *L1* showed the highest (78.2%) similarity to CPV4 and phylogenetic analysis showed that CPV24 clustered with CPV4 and CPV16 suggesting CPV24 is the third species 2 *Chipapillomavirus* type identified in dogs. This is the third report of extensive pigmented plaques covering a high proportion of the skin. Both previous cases were caused CPV4 and, considering the high genetic similarity between CPV4 and CP24, infection by these CPV types may predispose to more severe clinical disease. In addition, as plaques caused by CPV16 appear more likely to progress to neoplasia, the detection of a species 2 *Chipapillomavirus* within a pigmented plaque may indicate the potential for more severe disease.

## 1. Introduction

Papillomaviruses (PVs) are well established to cause a variety of hyperplastic and neoplastic diseases in dogs [1]. Papillomaviruses are classified using the *ORF L1* with PVs within the same genera sharing over 60% nucleotide similarity, PVs in the same species over 70% similarity, and PVs of the same type over 90% similarity [2]. There are currently 23 different fully sequenced and classified canine papillomavirus (CPV) types that are divided into three genera. The *Lambdapapillomavirus* genus contains two CPV types while the *Taupapillomavirus* genus contains eight CPV types. The *Chipapillomavirus* genus includes 13 different CPV types that are subdivided into three species. The Chipapillomaviruses are important as they are thought to cause canine pigmented plaques. While the majority of canine pigmented plaques remain small and only of cosmetic concern, there are previous reports of dogs with extensive plaques as well as rare reports in which plaques have progressed to neoplasia [3,4,5].

In the present report, the full genome of a novel PV type was amplified from a pug dog that developed extensive pigmented plaques. The PV was designated CPV24 with sequence and phylogenetic analysis suggesting classification as the third species 2 *Chipapillomavirus* of dogs. This classification is important as evidence suggests the species 2 *Chipapillomaviruses* may cause more severe disease than the other *Chipapillomavirus* types that infect dogs.

## 2. Materials and Methods

### 2.1. Case Summary and Sample Collection

A 9-year-old spayed female pug dog presented due to multiple dark slightly raised skin plaques the were most numerous on the ventrum and medially on the distal limbs. These plaques progressed in number and size over the next two years until the dog had numerous up to 3 cm diameter plaques that extended up to 3 cm from the surface of the skin (Figure 1) present in most areas of the body including the head. One of the lesions was sampled for histology which allowed a diagnosis of pigmented plaque to be confirmed.

The dog was initially treated with once-daily application of imiquimod cream (Aldara, iNova Pharmaceuticals (Australia) Pty Ltd., Chatswood, NSW, Australia) on a one week on, one week off basis for 12 months. Due to the extensive lesions, the cream was only used on plaques in one defined area of the body at a time. The cream resulted in, at best, mild improvement and treatment was changed to tigilanol tiglate 5 mg/mL gel (QBiotics Group, Toowong, QLD Australia) that was administered twice 9 days apart. While the lesions seemed to become more flattened, skin irritation developed, and the therapy was discontinued. The plaques continued to increase slowly over the following six months. At this time, the plaques started to cause significant discomfort and laser excision was used to remove some of the larger lesions. Excised samples were either fixed in formalin for histology or stored in saline for the determination of the causative PV.

### 2.2. Initial PCR and DNA Sequencing

DNA was extracted from an unfixed sample of pigmented plaque (NucleoSpin DNA FFPE XS kit, Macherey-Nagel, Düren, Germany) and PV DNA was amplified using the MY09/11 and CP4/5 consensus primers as previously described [4]. DNA extracted from a canine oral mass that contained CPV17 was used as a positive control while no template DNA was added to the negative controls. The amplified DNA was sequenced and compared to other sequences in GenBank using BLAST (https://blast.ncbi.nlm.nih.gov/Blast.cgi, accessed 1 May 2022).

The formalin-fixed sample was processed for histology. Examination revealed microscopic features consistent with a pigmented plaque. To ensure that DNA was extracted from the lesion, a sample from the centre of a pigmented plaque was taken from the tissue block using a scalpel. DNA was extracted and PV DNA was amplified from this sample using the same methods as before.

Taking advantage of the circular nature of the PV genome, the full genomic sequence was amplified by ‘outward facing’ primers that were designed using Geneious 10.2.6 based on the short sequence of *ORF L1* DNA amplified by the MY09/11 primers. The primers (CPV24L1F 5′- TCTCTTGCAACCCCATGTCC and CPV24L1R 5′-GCCGGACGTAGGTCTTGAAA; Integrated DNA Technologies, IA, USA) amplified an approximately 7300 bp region of DNA extracted from the unfixed sample. Amplification was performed using repliQa HiFi ToughMix (Quantabio, Beverly, MA, USA) according to the manufacturer’s instructions using an Eppendorf Mastercycler Nexus G2 (Hamburg, Germany). An Illumina sequencing library was prepared from the resulting PCR products using the Nextera XT DNA Library Preparation Kit (Illumina Inc., San Diego, CA, USA). Paired-end 2 × 250 bp sequencing of the DNA library was then performed on an Illumina MiSeq sequencer. The quality of the reads was confirmed using FastQC version 0.11.6. To assemble the viral genome, around 2.6 million reads were assembled into a single contiguous sequence using Spades 3.15 (accessed 1 August 2022) and Geneious 10.2.6 (Biomatters, Inc, Auckland, New Zealand) [6].

### 2.3. DNA and Protein Sequence Analysis

As the novel PV showed the greatest similarity to CPV4, the characteristics of the putative viral genes, the presence of conserved protein domains and motifs, and the presence of regulatory sequences were predicted by comparison to this papillomavirus type. This was done using the annotate from database tool in Geneious v10.2.6 in combination with Interpro (http://www.ebi.ac.uk/interpro/, accessed 1 August 2022). Genbank accession EF584537 was used as the reference sequence for CPV4.

### 2.4. Phylogenetic Analysis

Complete genomes of 84 PV representative types from each of the currently recognised genera were obtained from GenBank. Nucleotide sequences for the *ORF L1* were extracted and individually aligned using MAFFT in Geneious v10.2.6 resulting in an alignment of 1476 nucleotides [7]. Maximum likelihood analysis was performed using PhyML version 3.0 [8], available on the ATGC bioinformatics platform (http://www.atgc-montpellier.fr/phyml/, accessed 1 August 2022). Phylogenetic trees were inferred employing Subtree Pruning and Regrafting branch-swapping and nucleotide substitution models determined by Smart Model Selection [9]. Branch support was assessed using an approximate likelihood ratio test (aLRT) with the Shimodaira–Hasegawa-like procedure. The tree was produced using a general time-reversible model with invariable sites and gamma distribution (GTR + G [0.800] + I [0.115]). Tree visualization and annotation was performed with Evolview v3 [10]. Pairwise sequence similarities were calculated from the alignment of the complete *ORF L1* of the novel PV with that of other PV types.

### 2.5. Nucleotide Sequence Accession Number

The sequence of the novel PV was deposited in GenBank under accession number OP432240.

## 3. Results

### 3.1. Initial PV DNA Amplification

Papillomaviral DNA was amplified from DNA extracted from the unfixed sample of pigmented plaque using the MY09/11 and CP4/5 consensus primers. Comparison of the 357 bp section of the *ORF L1* revealed 100% similarity to a partial PV sequence previously submitted (GenBank JQ040500). The partial L1 sequence was 81% similar to CPV4. Comparison of the amplified sequence of the *ORF E1* revealed CPV4 was most similar, although the two sequences were only 76.4% similar over a 270 bp section of DNA. Identical sequences were amplified by the MY09/11 and the CP4/5 primers from DNA extracted from the sample taken from the histology tissue block. No PV DNA was amplified from any of the negative controls.

### 3.2. CPV24 Complete Gene Sequence

The complete genome of the PV was 7742 bp, with a GC content of 52.8%. The first nucleotide in the *ORF E6* was assigned number 1 in the sequence. As this is the 24th PV type detected in domestic dogs, it was designated CPV24.

### 3.3. Open Reading Frame Organization of CPV24 Genes

The PV was predicted to contain seven ORFs that coded for five early genes (E1, E2, E4, E6, E7) and two late genes (L1, L2; Figure 2). The predicted ORFs and characteristics of their putative protein products are shown in Table 1.

The PV oncoproteins E6 and E7 bind to host cell proteins, reactivating cell proliferation, modulating cell differentiation and enhancing cell survival [11,12]. The putative E6 protein of CPV24 consists of 151 amino acids (aa) and contains two conserved zinc-binding domains (CXXC-X29-CXXC) between aa 25–61 and 98–134, separated by 36 aa. The E6 sequence did not contain a PDZ-binding motif (ETQL) in its C-terminus. The 99 aa E7 protein also contained one conserved zinc-binding domain between aa 53–89 and a retinoblastoma (pRb) protein-binding site (LXCXE) at aa 23–27 [13].

The E1 and E2 proteins are required for efficient viral DNA replication and transcription. The predicted E1 protein was 629 aa, with N-terminal (aa 7–124) and C-terminal (aa 333–620) ATP-dependent helicase domains. The C-terminal domain contained the conserved ATP-binding site (GPPNTGKS) for the helicase at aa 457–464 and the RBD-like origin of the replication domain at aa 160–307. The binding of cyclin/cyclin-dependent kinase complexes to E1 is required for the initiation of PV DNA replication [14] and the predicted E1 protein had three such cyclin A interaction sites (RXL) at aa 73–75, 111–113 and 557–559. The E2 protein was 485 aa in length, consisting of an N-terminal transactivation helicase domain (aa 1–195) and a C-terminal viral DNA binding domain (aa 407–480). Unlike some other PV types, there was no leucine-zipper domain (LX6LX6LX6L) in the putative CPV24E2 protein [15].

The putative *ORF E4* was present within the *ORF E2* region, but in a different translation frame. Like E1 and E2, E4 contributes to the success of viral DNA amplification [16]. A signal peptide was predicted at aa 1–31, and a non-cytoplasmic domain at aa 32–288. The CPV24 E4 protein does not have a high proline content (12.5%), although as occurs in other PV E4 proteins [16], two proline rich domains, between aa 130–147 and aa 228–241, were present.

The late region encodes two viral capsid proteins, L1 (501 aa) and L2 (506 aa). Both proteins contain a high proportion of positively charged residues (K and R) in the C-terminal end, thought to be involved in interactions with heparan sulfate during entry to the cell. The highly conserved Y-R dipeptide motif was present in L1 at aa 415–416, and this motif is possibly associated with canyon wall binding to heparan sulfate [17]. The predicted L2 protein also contains the conserved N-terminus furin cleavage motif between aa 6–9 (RXK/R-R) and the conserved C-terminus L1-binding site at aa 467–470 (PXXP motif). Furin cleavage is required for internalization of the virion into the host cell [18].

The long control region (LCR) encompasses 540 bp (nt 7203–7742) between L1 and E6. This region does not code for a protein but is important for regulation of viral gene transcription. The CPV24 LCR contains the expected E1 and E2 binding sites (E1BS and E2BS). The E1BS sequence motif is reported to be a hexanucleotide palindromic sequence, 5′-AT(A/G/T)G(C/T)(C/T)-3′ separated by 3 nucleotides [19]. An E1BS was predicted at nt positions 7625-7639, with the following sequence: 5′-ATTGTTGCTGACAAC-3. Four putative E2BS, with a consensus sequence ACCN6GGT, were identified at nt 7319, 7498, 7587, and 7671. The position of the E2BS at nt 7498, 7587 and 7671 resulted in the expected flanking of the E1BS, as is typical for the PV origin of replication [20]. Adjacent to this area in the LCR was the TATA box at nt 7651.

### 3.4. Phylogenetic Analysis of CPV24

The maximum likelihood tree comparing CPV24 with other PV types from GenBank confirmed their grouping within the *Chipapillomavirus* genus (Figure 3). CPV24 clustered with the species 2 *Chipapillomaviruses* (CPV4 and CPV16) with high branch support.

### 3.5. CPV24 Sequence Similarity to Other Papillomaviruses

The *ORF L1* nucleotide sequence of CPV24 has 78.2% similarity to CPV4 and 68.0% similarity to CPV16 (Table 2). CPV24 has between 65.6% (CPV5) and 69.7% (CPV12) similarity to the species 1 *Chipapillomaviruses* and between 61.9% (CPV14) and 64.2 (CPV8) to the species 3 *Chipapillomaviruses*. When compared other CPV types, CPV24 has 56.1% similarity to CPV1 (a *Lambdapapillomavirus*) and 54.1% similarity to CPV2 (a *Taupapillomavirus*). Comparison of CPV24 to PVs within other genera revealed the most similarity with a porcine *Dyodeltapapillomavirus* (60.3%) and a feline *Dyothetapapillomavirus* (59.6%).

## 4. Discussion

The detection of CPV24 DNA in the pigmented plaque suggests that this PV could have caused lesion development. However, proving causality is difficult as PVs frequently asymptomatically infect skin [21]. In the present case, CPV24 DNA was amplified from a sample that was known to be from the centre of a pigmented plaque. However, while this proves CPV24 was present within the lesion and not simply present as an incidental infection of the surrounding epidermis, it does not confirm the PV caused the pigmented plaque. Further evidence supporting a causal role of CPV24 was derived from the failure to amplify other PV types from the plaques. Additionally, CPV24 is closely related to CPV4 which has previously been associated with pigmented plaque development in multiple studies [1,22].

Comparison of the CPV24 *ORF L1* to that of other PVs suggests the novel virus is a *Chipapillomavirus* type. As CPV24 was much more similar to CPV4 than any other PV type and, as CPV24 clustered phylogenetically with CPV4 and CPV16, it is proposed that CPV24 is classified as the third series 2 *Chipapillomavirus* of dogs.

The skin lesions associated with CPV24 in the presently described pug dog were unusually severe with the plaques large, exophytic and covering a high proportion of the skin. There are two previous reports of similar extensive plaques in dogs [3,4]. These were reported in a golden retriever and in a chihuahua and plaques in both dogs were caused by the closely related CPV4. While additional cases need to be evaluated, this suggests that pigmented plaques caused by either CPV4 or CPV24 may be more extensive than plaques caused by the other *Chipapillomavirus* types. Additionally, another species 2 *Chipapillomavirus*, CPV16, is over-represented in the small number of pigmented plaques that have been reported to progress to neoplasia [5,23]. Therefore, the detection of a species 2 *Chipapillomavirus* in a canine pigmented plaque may indicate a higher potential for disease progression and a worse prognosis.

While the pathogenesis of canine pigmented plaques is not fully resolved, this disease has similarities to the human condition epidermodysplasia verruciformis (EV). EV is caused by an inherited or acquired immune defect that prevents the development of a normal immune response against cutaneous PV infection [24]. As seen in some humans with EV, the present dog did not show other evidence of immune deficiency suggesting that, if the lesions developed due to an immune dysfunction, only the immunity against PV infection was affected.

Since the role of PVs in the development of canine pigmented plaques was first recognized in 2004 [25], numerous plaques have been evaluated for the presence of PV DNA. The evaluation of these plaques allowed 13 different CPVs to be fully sequenced. However, prior to the present report, CPV24 DNA had only been detected once when a short sequence of novel PV DNA was reported in one of a series of 27 plaques [26]. Therefore, CPV24 appears to be a rare cause of canine pigmented plaques.

Further research is required to determine the mechanisms by which CPV24 infection influences cell growth and causes lesion development. However, the putative CPV24 E7 protein has a pRb binding site which could prevent normal regulation of cell growth and differentiation [12]. In addition, the putative CPV24 E6 contains two conserved zinc-binding domains that may allow the protein interactions that promote replication of infected cells. As pigmented plaques rarely progress to cancer, most *Chipapillomavirus* types appear unable to disrupt cell regulation sufficiently to cause neoplastic transformation. However, CPV16 may more strongly predispose to neoplasia, at least in part, due to the ability of this PV type to become integrated with the cell DNA [23,27]. While both CPV16 and CPV24 are species 2 *Chipapillomaviruses,* it is unknown whether CPV24 is also predisposed to accidental integration and therefore causes the progression of a plaque to neoplasia.

## 5. Conclusions

CPV24 is the second CPV type to be associated with the development of extensive pigmented plaques in dogs. This PV is the third species 2 *Chipapillomavirus* type identified in dogs. There is some evidence that the species 2 *Chipapillomaviruses* may cause more severe disease than the other *Chipapillomaviruses* that infect dogs. Therefore, the detection of CPV24 within a pigmented plaque could indicate that plaque progression is more likely.

## Figures and Tables

**Figure 1 viruses-14-02357-f001:**
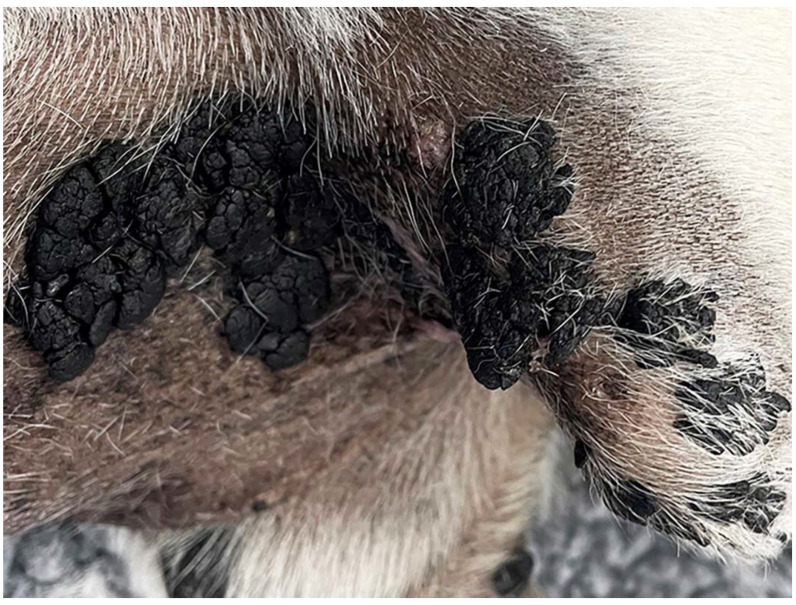
Large exophytic pigmented plaques developed over a wide area of the 9-year-old pug dog.

**Figure 2 viruses-14-02357-f002:**
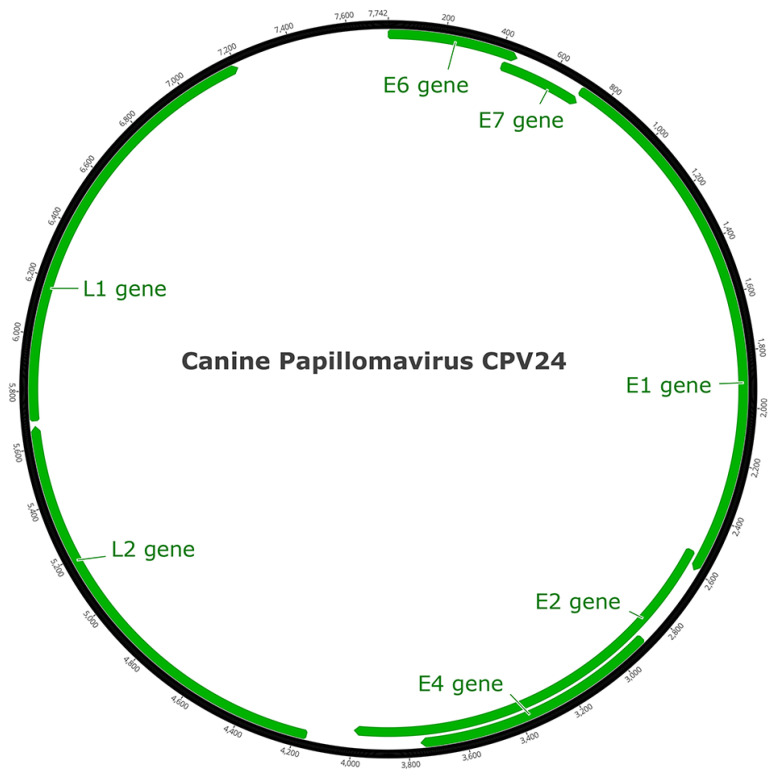
Schematic genomic organization of CPV24.

**Figure 3 viruses-14-02357-f003:**
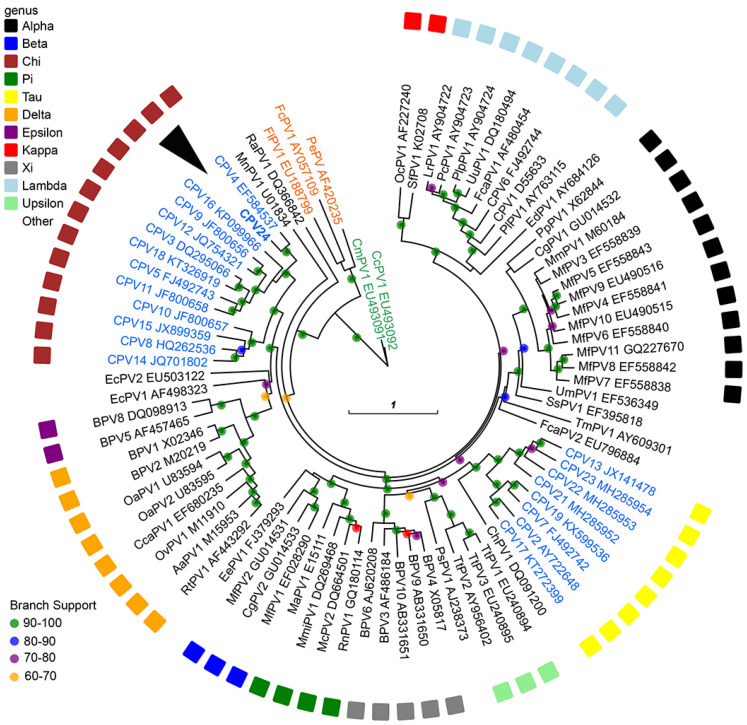
Unrooted Maximum likelihood phylogeny based on concatenated nucleotide alignment of E1, E2, L1 and L2 ORF sequences from putative CPV24 (arrowhead) with 84 other PV types of different species and genera. Accession numbers for the sequences used are included. Abbreviations used include Micromys minutus papillomavirus, MmiPV; Mastomys coucha papillomavirus, McPV; Canine papillomavirus, CPV; Bovine papillomavirus, BPV; Bettongia penicillata papillomavirus, BpPV; Macaca fascicularis papillomavirus, MfPV; Felis catus papillomavirus, FcaPV; Equus caballus papillomavirus, EcPV; Multimammate rat papillomavirus, MnPV; Psittacus erithacus timneh papillomavirus, PePV; Fringilla coelebs papillomavirus, FcPV; Francolinus leucoscepus papillomavirus, FlPV; Ovis aries papillomavirus, OaPV; Oryctolagus cuniculus papillomavirus, OcPV; Sylvilagus floridanus papillomavirus, SfPV; Rousettus aegyptiacus papillomavirus, RaPV; Capreolus capreolus papillomavirus, CcaPV; Odocoileus virginianus papillomavirus, OvPV; Alces alces papillomavirus, AaPV; Rangifer tarandus papillomavirus, RtPV; Erinaceus europaeus papillomavirus, EePV; Colobus guereza papillomavirus, CgPV; Mesocricetus auratus papillomavirus, MaPV; Rattus norvegicus papillomavirus, RnPV, Phocoena spinipinnis papillomavirus, PsPV; Tursiops truncatus papillomavirus. TtPV; Capra hircus papillomavirus, ChPV; Trichechus manatus latirostris papillomavirus. TmPV; Sus scrofa papillomavirus. SsPV; Ursus maritimus papillomavirus, UmPV; Macaca mulata papillomavirus. MmPV; Pan paniscus papillomavirus, PpPV, Erethizon dorsatum papillomavirus. EdPV; Procyon lotor papillomavirus, PiPV; Uncia uncia papillomavirus, UuPV; Panthera leo persica papillomavirus, PlpPV; Puma concolor papillomavirus. PcPV; Lynx rufus papillomavirus, LrPV. The PV genera are also listed. Internal branches are coloured based on inferred branch support values, as determined by 1000 replicates usin RAxML. The scale bar indicates the genetic distance (nucleotide substitutions per site).

**Table 1 viruses-14-02357-t001:** Predicted ORFs in the CPV24 genome. pI indicates the isoelectric point.

ORF	ORF Location	Length (nt)	Length (aa)	Molecular Mass (kDa)	pI
E1	705–2594	1890	629	71.27	5.48
E2	2536–3993	1458	485	53.85	6.69
E4	2891–3757	867	228	31.6	7.61
E6	1–456	456	151	17.33	8.32
E7	416–715	300	99	11.16	4.2
L1	5697–7202	1506	501	56.18	8.15
L2	4158–5678	1521	506	53.68	5.91

**Table 2 viruses-14-02357-t002:** Percent identity between the proposed CPV24 and other papillomaviruses based on the pairwise nucleotide alignments of the papillomavirus *ORF L1*. The alignments were performed using MAFFT in Geneious v10.2.6 using default parameters.

Papillomavirus	Host Species	Classification	L1 Similarity (%)
Canine familiaris papillomavirus 4 (EF584537)	Domestic dog	Chipapillomavirus 2	78.2
Canine familiaris papillomavirus 12 (JQ754321)	Domestic dog	Chipapillomavirus 1	69.7
Canine familiaris papillomavirus 9 (JF800656)	Domestic dog	Chipapillomavirus 3	69.0
Canine familiaris papillomavirus 16 (KP099966)	Domestic dog	Chipapillomavirus 2	68.0
Canine familiaris papillomavirus 3 (DQ295066)	Domestic dog	Chipapillomavirus 1	67.5
Sus scrofa papillomavirus 1 (EF395818)	Domestic pig	Dyodeltapapillomavirus	60.3
Felis catis papillomavirus 2 (EU796884)	Domestic cat	Dyothetapapillomavirus 1	59.6
Canine familiaris papillomavirus 1 (D55633)	Domestic dog	Lambdapapillomavirus	56.1
Canine familiaris papillomavirus 2 (AY722648)	Domestic dog	Taupapillomavirus	54.1

## Data Availability

Not applicable.

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
