# Peer review of "Genomic Characterisation of Canis Familiaris Papillomavirus Type 24, a Novel Papillomavirus Associated with Extensive Pigmented Plaque Formation in a Pug Dog"

_viruses, 2022, doi:10.3390/v14112357_

Round 1
Reviewer 1 Report
I believe that Munday et al paper is a continuation of the previous paper (Munday, J.S.; Lam, A.T.H.; Sakai, M. Extensive progressive pigmented viral plaques in a Chihuahua dog. Vet Dermatol 2022, 33, 288 252-254.). In fact it seems rather strange that the breed of the dog in which the lesions were found is not specified and that there are no images of the whole dog. I believe that if my assumption is correct, it must be made clear in the paper.
Minor revision: describe the paragraph 2.3. DNA and protein sequence analysis more clearly and in detail.
Author Response
I believe that Munday et al paper is a continuation of the previous paper (Munday, J.S.; Lam, A.T.H.; Sakai, M. Extensive progressive pigmented viral plaques in a Chihuahua dog. Vet Dermatol 2022, 33, 288 252-254.). In fact it seems rather strange that the breed of the dog in which the lesions were found is not specified and that there are no images of the whole dog. I believe that if my assumption is correct, it must be made clear in the paper.
The two papers describe two different cases. The earlier paper in Vet Dermatology described a 6-year-old Chihuahua dog. Plaques in the Chihuahua were found to be caused by CPV4 using PCR. The previous case was important due to the unusual breed and the extensive plaques. In the present case, the plaques developed on a 9-year-old pug dog. PCR revealed a previously-undetected PV type which is why this case is important. Although the breed of the dog in the present case was stated in the title, abstract and line 51, that the two cases were separate was obviously not clear enough in the manuscript and the manuscript has been changed to clarify this (lines 43 and lines 241-245).
The authors agree it is a shame no images of the whole dog were included in the manuscript. However, unfortunately no suitable images of this dog were taken with Figure 1 the best to demonstrate the proliferative nature of the plaques.
Minor revision: describe the paragraph 2.3. DNA and protein sequence analysis more clearly and in detail.
This paragraph has been rewritten to describe more clearly what was done and to provide greater detail as suggested (lines 98-103).
Reviewer 2 Report
in this report, the authors describe the isolation of a novel canine papillomavirus from large exophytic plaques that developed on a 9-year-old pug. Since this is the 24th papillomavirus isolated from dogs, they named it CPV24. CPV24 is most closely related to CPV4, which has been associated with similar lesions. Overall the study is straightforward and well done and the technical procedures are adequately described. This is an interesting addition to papillomavirus literature.
Author Response
Thank you for your positive review.